# In Search of the Perfect Triple BB Bond: Mechanical Tuning of the Host Molecular Trap for the Triple Bond B≡B Fragment

**DOI:** 10.3390/molecules26216428

**Published:** 2021-10-25

**Authors:** Shmuel Zilberg, Michael Zinigrad

**Affiliations:** Materials Research Center, Department of Chemical Sciences, Ariel University, Ariel 40700, Israel; zinigradm@ariel.ac.il

**Keywords:** B≡B triple bond, diboryne, π-back-donation, host molecule, strain energy

## Abstract

The coordination of the B_2_ fragment by two σ-donor ligands L: could lead to a diboryne compound with a formal triple bond L:→B≡B←:L. σ-Type coordination L:→B leads to an excess of electrons around the B_2_ central fragment, whereas π-back-donation from the B≡B moiety to ligand L has a compensation effect. Coordination of the σ-donor and π-acceptor ligand is accompanied by the lowering of the BB bond order. Here, we propose a new approach to obtain the perfect triple BB bond through the incorporation of the BB unit into a rigid molecular capsule. The idea is the replacement of π-back-donation, as the principal stabilization factor in the linear NBBN structure, with the mechanical stabilization of the BB fragment in the inert molecular capsule, thus preserving the perfect B≡B triple bond. Quantum-chemical calculations show that the rigid molecular capsule provided a linear NBBN structure and an unusually short BB bond of 1.36 Å. Quantum-chemical calculations of the proposed diboryne adducts show a perfect triple bond B≡B without π-back-donation from the B_2_ unit to the host molecule. Two mechanisms were tested for the molecular design of a diboryne adduct with a perfect B≡B triple bond: the elimination of π-back-donation and the construction of a suitable molecular trap for the encapsulation of the B_2_ unit. The second factor that could lead to the strengthening or stretching of a selected chemical bond is molecular strain produced by the rigid molecular host capsule, as was shown for B≡B and for C≡C triple bonds. Different derivatives of icosane host molecules exhibited variation in BB bond length and the corresponding frequency of the BB stretch. On the other hand, this group of molecules shows a perfect triple BB bond character and they all possess a similar level of HOMO.

## 1. Introduction

For the last decade, the boron–boron (B≡B) triple bond has been at the center of a dispute [1,2]. Braunschweig proposed [2] the chemical evidence of the BB triple bond’s character: diboryne was found to react with chalcogens, affording the [2.2.1]-bicyclic systems via a six-electron process involving the insertion of five chalcogen atoms into the BB triple bond, which was completely cleaved during the reaction. This widely discussed issue is an example of a successful case study acquiring common interest [3]. The difficulty of this discussion is the absence of an obvious reference point—the parent molecule with a perfect B≡B triple bond. For example, Köppe and Schnöckel suggested that the BB bond in the diboryne adduct is intermediate between a single and a double bond [4].

Compounds with B-B multiple bonds are still rare [2]. Formally, the coordination of the B_2_ fragment by two σ-donors L: should produce the structure with a triple bond L:→B≡B←:L (I) between B-atoms. The trivial Lewis structure hides the fact that the ground state of the parent B_2_ species is a triplet state ^3^Σ_g_^−^ with a pair of degenerate π-MO populated by two electrons with parallel spin: (σ^+^)^2^(σ^−^)^2^(π_x_)^1^(π_y_) [1,5]. The electronic configuration with full π-MO: (σ^+^)^2^(σ^−^)^0^(π_x_)^2^(π_y_)^2^ is a high-lying excited singlet state ^1^Σg+ [6,7]. The singlet state of B_2_−^1^Σg+ has a triple bond electronic configuration involving one σ-bond, two π-bonds, and two empty sp-hybrid orbitals at both B-atoms (see the MO interpretation in Ref. [8]). Consequently, the structure with the B≡B triple bond requires very strong L:→B coordination for the compensation of the preference of the triplet electronic configuration of the parent B_2_ moiety. The dicarbonyl adduct OCBBCO was the first experimentally detected diboryne. It was prepared in an argon matrix at 8 K, characterized by IR spectrum, and backed by quantum-chemical calculations [9,10]. The Lewis structure with a triple BB bond–OC:→B≡B←:CO for a singlet ground state was proposed by Zhou et al. [9,10]. The second representative of the diboryne family, the diboronyl diboryne anion OBBBBO^−1^ (bond order 2.5), was detected and investigated in a gas phase using photoelectron spectroscopy [11]. Bond order analyses of dianion OBBBBO^−2^ concluded the presence of a true triple B≡B bond in this negatively charged complex [11]. Later, a compound with a CBBC linear fragment was synthesized by Braunschweig et al. [12]. This compound, which is stable at a room temperature, combines two N-heterocyclic carbene (NHC) ligands coordinated to the B_2_ fragment. The conclusion about the B≡B triple bond’s character was based on the X-ray data, according to which the central CBBC fragment is linear and the BB distance 1.449 Å is considerably shorter, by ~0.1 Å, than the corresponding value for the double B=B bond. DFT calculations [1,12,13,14] supported the findings regarding the structural characteristics, such as the linearity of the CBBC fragment and R(BB) < 1.46 Å. Calculations show the occurrence of the two π-HOMOs localized on the BB fragment, which, on the MO level, indicates the triple B≡B bond’s character [1,13]. Different symmetric adducts L→B≡B←L (I, L= CO, CS, N_2_) were computationally studied by Mavridis et al. [15] and later by Frenking et al. [8,13,14]. The bonding scheme for the complexes with an unsaturated ligand was interpreted in terms of donor–acceptor interactions between the σ-donor, π-acceptor ligands L and B_2_ as the σ-acceptor, and the π-donor moiety [13,14,15]. Frenking et al. [14] specified that the π-back-donation L→ B≡B ←L for L=CO, N_2_ is very strong, but this is still a triple bond. Braunschweig et al. [12] also supported the interpretation of diboryne (I) as a structure with a B≡B triple bond. Later, the NHC-diboryne adduct was investigated by Raman spectroscopy [16] and the Raman active BB stretching mode was observed at 1653 cm^−1^, which is in agreement with the B3LYP predictions of 1681 cm^−1^. 

We attempted to find the answer to the problem concerning the perfect B≡B bond through the elimination of the π-back-donation effect, which is the dominant factor in the weakening of the π-components of the triple bond. On the other hand, the design of such a molecule must compensate for the loss of π-back-donation stabilization and the need to remain linear. 

## 2. Results and Discussion

Experimental observation of a stable compound with a formal B≡B triple bond initiated the discussion about the physical indication of this bond. These included the following: an assessment of the structural characteristics, such as the linearity of the central LBBL fragment and BB bond length [1,2,3,4,13,14]; experimental and computational assessments of the strengths of the BB triple bond through the stretching frequencies of their central BB unit [16]; a qualitative depiction of the charge transfer channels (σ, π, and polarizations)
[17]; advanced solid-state NMR and computational methodology were used in order to directly experimentally probe the orbitals involved in multiple boron–boron bonds via the analysis of ^11^B–^11^B spin–spin (*J*) coupling constants [18]. Köppe and Schnöckel contend that the force constant of the BB bond is lower than expected for a B≡B triple bond and the bond order is only slightly larger than 1.5; thus, consequently, NHC diboryne “does not contain a BB triple bond” [4]. Other authors rejected this conclusion [1,3,18,19].

The results of the previous studies of diboryne adducts L→ B≡B ←L can be summarized as follows (Table 1): The numerous calculated data and two experimentally studied diborynes show: (1) the linearity of the LBBL fragment is caused by the maximal overlap of the empty σ-orbital of the B_2_ fragment with the σ-lone pair of the ligand and the effective back-donation to the π-system of ligands. Trans-bending distortion leads to the loss of π-bonding [20,21]. VB calculations also show that the σ-frame favors trans distortion, while the π-system opposes it [22]; (2) diboryne’s BB bond is 1.44 ÷ 1.47 Å, which agrees with the standard value for a triple bond (1.46°A) [23], and it is shorter than the B=B double bond by >0.1 Å [17]; (3) the experimentally observed and computationally predicted Raman-active BB stretch in diboryne is near to 1700 cm^−1^ [16].

Strong π-back-donation is an obvious reason for the stabilization of a linear LBBL structure; however, on the other hand, this mechanism is responsible for the lowering of π-bonding between boron–boron atoms. Frenking summarized it as follows: “Thus, the bond order for the B–B can be expected between 2 and 3 while the triple bond character is retained in the diboryne whose bonding situation is properly sketched with the formula NHC→ B≡B ←NHC” [1]. Noble gas diborynes NG-BB-NG (NG = Ar and Kr) [15] are examples of σ-adducts without π-back-donation from B_2_ to ligands L. The linear structure of NG-BB-NG Ar and Kr diborynes has a short < 1.4 Å BB bond (Table 1), but it undergoes spontaneous distortion into a zigzag configuration that is a direct consequence of the loss of the π-back effect.

The model compound bis-1-azaadamantane-diboryne adduct C_9_H_15_N→BB←NC_9_H_15_ (II) is a non-rigid molecule with a shallow potential well corresponding to the two slightly distorted zigzag configurations (II). The configuration with a linear NBBN central fragment is a transition structure (II-TS, the barrier height of 1kcal/mol) that has a short BB bond length R(BB) = 1.422 Å (Figure 1).

### 2.1. Molecular Design of Compounds with the Perfect Triple B≡B Bond

According to a recent suggestion [24], the perfect B≡B triple bond can be formed by σ-coordination of the N-atom with the B_2_ fragment, provided that the π-back-donation is prevented. We propose a new approach for designing compounds with the perfect triple B≡B bond—the incorporation of the B_2_ unit in a rigid saturated host structure with two nitrogen σ-donor centers. There is no π-acceptor in such an inert saturated host molecule, and, consequently, the π-back-donation effect is absent. Coordination of the B_2_ unit by N-atoms acts as the stabilization factor, whereas the rigid saturated frame ensures the linearity of the NBBN central fragment. Cryptand-type host molecules (HM) look suitable candidates for this. Host molecules (III)-(VII) and corresponding diboryne adducts III-B_2_ ÷ VII-B_2_ were studied at the DFT level—B3LYP [25,26] and M06-2X [27]. All structures were optimized, and harmonic vibrational frequencies were calculated. GAUSSIAN-09 [28] and GAMESS [29,30,31,32] quantum chemical program suits were used. 

Three types (Figure 1) of host molecules with bicyclic saturated frames were used for the compututational design of B_2_ trap-Cryptand 222-4,7,13,16,21,24-hexaoxa-1,10-diazabicyclo[8.8.8]hexacosane (III); 1,8-diazabicyclo[6.6.6]icosane (IV); 3,6,10,13,16,19-hexaoxa-1,8-diazabicyclo[6.6.6]icosane (V).

Host icosane molecules IV and V were modified by two terminal adamantane units—structures VI and VII—for the compression of the central icosane part (Figure 2).

DFT calculations (M06-2X, Table 2) show that diaza-compounds III-V (Figure 1) and their adamantane derivatives VI and VII (Figure 2) produce stable B_2_-host complexes III-B_2_÷VII-B_2_. Polycyclic diboryne adducts III-B_2_ ÷ VII-B_2_ have a linear NBBN fragment (calculations show very slight bending at the boron atoms NBB > 178.5°) and a short BB bond. The NBO (natural bond orbital) [33] analysis implemented in the GAUSSIAN package [28] shows a dominant (>99%) Lewis structure with a B≡B triple bond and a single NB bond. A comparison of diboryne complexes III-B_2_–VII-B_2_ with the reference diaza-adduct N_2_BBN_2_ shows the expected effects, namely, π-back-donation strengthens the N-B bond and simultaneously weakens the BB triple bond (Table 2). Diaza complex N≡N:→B≡B←:N≡N has an obvious π-back-donation effect, but this feature is absent in the molecules III-B_2_ ÷ VII-B_2_. Calculations show an identical picture: one σ-MO and two π-MOs between two boron atoms for all adducts III-B2 ÷ VII-B2. However, the BB bond lengths (Table 2) are shortened in the series III > IV > VI > V > VII despite the similar orbital structure. It must be noted that the experimentally observed BB bond lengths for (NHC)BB(NHC) [12] are longer and weaker (ν(BB) < 1700 cm^−1^) than the calculated BB distance and the corresponding BB frequencies for all studied adducts III-B_2_ ÷ VII-B_2_ (Table 2). The variation in the N-B bond distance is stronger than the change in the B-B bond length in all cases. This is an obvious consequence of the different bond strength, namely, the BB bond is a very strong triple bond whereas NB is an ordinary single bond.

### 2.2. Strain Energy of the Host Molecules

Host molecules III, IV, and V have different inner cavity sizes for complexation with B_2_, but the complexation of B_2_ is effective for all diaza hosts III-VII. The structural/mechanical characteristics of the host molecules are as follows: the icosane derivatives IV and V have a small frame with a six atom edge, whereas the hexacosane cryptand-222 (III) has a bigger capsule with an eight atom edge. Comparison between the initial diaza-host III-VII compound and the corresponding host-diboryne adduct shows the trivial linkage between two structural parameters, i.e., the shorter N^…^N distance of the host molecule leads to a shorter BB bond for the diboryne adduct. In other words, the strain of the molecular capsule determines the level of the compression or stretch of the BB fragment. For example, cryptand-222 (III) has the longest NN distance and adduct III-B_2_ exhibits the maximal length BB bond relative to the other studied complexes. On the other hand, VII-hexa-oxo-icosane framed by two adamantane units has the shortest NN distance, i.e., only 3.638 Å, and this is a perfect host molecule for the compression of the incorporated BB unit. VII-B_2_ has an extremally short (1.347 Å) and strong (ν(BB)=2091 cm^−1^) BB bond. π-Back-donation is absent in both adducts III-B_2_ and VII-B_2_ and they differ only in terms of the mechanical behavior of the molecular frame. Host molecules are strongly distorted by incorporation of the BB fragment, which produces strain.

We estimated the strain energy of the host molecule by comparing the optimized structure with the structure of the host trap from the optimized diboryne adduct (∆∆E (host strain)). The large host molecule cryptand-222 (III) shows significant strain in its diboryne adduct III-B_2_ (∆∆E (host strain) = 40.3 kcal/mol). The cavities of icosanes IV-VI are more appropriate for the incorporation of the B_2_ fragment and the strain energy of the host capsule is reduced to ~32–33 kcal/mol. However, the strain of the host molecule VII with the smallest cavity (R(N^…^N) = 3.638 Å only for the free host molecule) reaches the highest strain level of 42.8 kcal/mol in its strongly compressed adduct VII-B_2_ (Table 2).

The addition of B_2_ to the host molecule III leads to the shortening of the NN distance from 6.048 Å to 4.543 Å. This deviation from the optimal cavity size produces strain, which could lead to the increased stretch of the BB and NB bonds. This is a reason for the BB bond length in the diboryne of hexacosane cryptand-222 (III-B_2_) being longer and weaker than in other diboryne complexes IV-VII, despite the perfect triple bond electronic characteristics. The frequency of the BB stretch of 1710 cm^−1^ (III-B_2_) is also the lowest in the group (Table 2). On the other hand, the distance between the two N-atoms in the parent host molecule VII is shorter than in its diboryne adduct VII-B_2_. This means that the driving force of the host molecule VII is a compression of the BB fragment.

The BB bond length is shorter than 1.4 Å for all icosane diborynes, but it is 1.347 Å for VII-B_2_, which is 0.1 Å shorter than the BB bond in the experimentally detected carbene-diboryne adduct [12,16].

Previous calculations revealed the symmetric Raman-active BB mode, which could serve as an indicator for a multiple BB bond [16]. The frequency of this mode for all the studied linear neutral L-BB-L molecules varies from 1720 to 1800 cm^−1^ [24]. Strengthening of the central BB fragment was accompanied by a strong redshift of up to 2092 cm^−1^ of the corresponding BB stretch mode for VII-B_2_ (Table 2, frequencies ν(BB)) in the series of studied polycyclic borynes. This is the pure BB stretch mode in all studied cases III-B_2_–VII-B_2_. The Raman-active stretching mode of the triple N≡N bond in molecular nitrogen N_2_ is observed at ∼2300 cm^−1^ [34]. The stretching frequencies of the C≡C bond in alkynes are normally in a range from ∼2100 to 2300 cm^−1^ [35]. A Raman-active B-B vibration at 2092 cm^−1^ (VII-B_2_, Table 2) indicates that the B_2_ fragment of diborynes III-B_2_–VII-B_2_ is a perfect B≡B triple bond on the same level as a classical N≡N and C≡C triple bonds. In addition, the experimentally observed [16] BB stretch at 1653 cm^−1^ indicated weaker BB bond compared to the corresponding feature in the compounds presented in our study (Table 2). This is consistent with a previous interpretation in which the bond order of (NHC)_2_ B_2_ is intermediate between a double B=B and triple B≡B bond [1,24].

DFT calculations show that the two-body dissociation of the diboryne adduct to ^3^B_2_ (ground state of B_2_) [5,6] and the corresponding diaza-polycycle is a strongly endothermic process ∆∆E(dissociation) ≥ 50.0 kcal/mol (Table 2):
^1^Host-B_2_ → ^1^Host + ^3^B_2_ (Host=III, IV, V, VI, VII)

The stability of the diboryne adducts III-B_2_–VII-B_2_ is provided by effective NB σ-coordination, while π-back stabilization is absent. Nevertheless, the diboryne derivatives of diaza-host molecules III-B_2_ –VII-B_2_ are thermodynamically stable compounds according to the M06-2X/cc-pVDZ calculations (Table 2). The dissociation energies of diboryne adducts III-B_2_-VII-B_2_ are approximately of the same order as for N_2_B_2_N_2_, which has strong π-back stabilization. Our attempts to detect the rupture of one NB bond along the antisymmetric NBBN stretch were unsuccessful despite the significant strain energy. These data could serve as an indication of kinetic stability.

Time dependent TD-M06-2X calculations of the Franck–Condon area provide a similar picture for all studied cases (III-B_2_–V-B_2_), namely, the lowest singlet excited state S_1_ is ~3.5 eV above the ground state S_0_, whereas the experimentally observed (NHC)_2_B_2_ adduct has a gap of 2.4 eV [12]. The lowest excited state is a triplet state ΔE(S_0_−T_1_) ≥ 1.5 eV, which has a zigzag structure for NBBN.

The conjugation of the π-donor B_2_ group with a ligand π-acceptor provides the lowering of the degenerate π-HOMO level, which is strongly localized on the BB fragment. Calculations of compounds III-B_2_ ÷ VII-B_2_ (Table 3) show an approximately equal HOMO level, which is 0.7 eV higher than in the case of (NHC)BB(NHC). The HOMO level in the experimentally observed (NHC)BB(NHC) compound must be lower relative to the case without π-back-donation. It must be noted that all these diboryne adducts III-B_2_ ÷ VII-B_2_ have different BB bond lengths despite the similar π-HOMO levels.

The IP (ionization potential) estimations, i.e., the calculated difference between the energy of the neutral and ionized form of the corresponding compounds in the ground state configuration, fully agree with the relative HOMO level. This is a direct indication on the very similar BB bonds in the group III-B_2_ ÷ VII-B_2_. Consequently, all the studied diboryne adducts of bicyclic host molecules III-VIII have a de facto perfect B≡B triple bond. Consequently, the differences in BB bond lengths between molecules in the group III-B_2_ ÷ VII-B_2_ do not have the electronic donor–acceptor origin. We proposed that the induced strain has a mechanical origin, because the host molecules III-VII differ in size, configuration, and distance between N-atoms in the cavity (see survey of the strained organic molecules [36]).

The mechanical hypothesis could be analyzed through a comparison of diboryne adducts III-B_2_ ÷ VII-B_2_ with analogue propellane molecules III-C_4_-VII-C_4_ with a C-C≡C-C fragment instead of a NBBN unit (Figure 2). No donor–acceptor interaction or π-back-donation occur in the case of pure carbon structures, which provides an opportunity to estimate the mechanical effect. Di-substituted alkynes VIII and IX (di-substituted acetylenes R- C≡C-R (R = 2-Adamantyl (VIII) and R=Methyl (IX))) are the reference triple bond C≡C molecules. The acetylene derivative III-C4 shows lengthening (0.015 Å) and weakening (blueshift Δν~−100 cm^−1^) of the C≡C bond relative to VIII and IX (Table 4). The contraction (0.02 Å) and strengthening (redshift Δν~150 cm^−1^) of the C≡C triple bond is unusual and is the result of a pure mechano-chemical compression effect. These results show the strong mechanical influence of the molecular frame on the C≡C triple bond.

The addition of an adamantane unit to the molecular frame IV or V leads to the compression of the central fragment B≡B or C≡C in all cases, i.e., IV-B_2_ vs. VI-B_2_ (Table 2), IV-C_4_ vs. VI-C_4_ (Table 4), V-B_2_ vs. VII-B_2_ (Table 2), and V-C_4_ vs. VII-C_4_ (Table 4). The effect is especially strong following the addition of the adamantane part to hexa-oxo-frame V-ΔR(BB) = 0.037 Å and Δν(BB) = 164 cm^−1^. This is an example of mechano-chemical compression leading to the amplification of the chemical bond.

## 3. Conclusions

Producing the perfect B≡B triple bond was shown to depend on two factors: (a) the elimination of π-back-donation, which means the perfect π-bonding can be preserved; and (b) the construction of the compressing molecular capsule, which allows the linear NBBN configuration to be maintained. Bicyclic host compounds IV-VII satisfy the following requirements: (a) the molecular capsules have a suitable size and configuration cavity; and (b) there are no π-acceptors in the host molecule, which excludes π-back-donation. The BB bond lengths of diboryne adducts in the selected host molecules are significantly shorter (R(BB) < 1.4 Å) and stronger (ν(BB) > 1850 cm^−1^) than those in the earlier experimentally detected diboryne derivatives.

The BB bond lengths and the frequencies of the Raman-active B-B mode indicate that the B_2_ fragment of diborynes III-B_2_–VII-B_2_ has a perfect B≡B triple bond character, as is the case in well-known classical N≡N and C≡C triple bonds.

The studied group of host molecules exhibits the mechanical strain effect on the triple BB bond lengths and corresponding BB frequencies without changing the BB bond order or the level of HOMO. An analogous molecular-mechanical effect was also detected for the corresponding compounds with a C≡C triple bond.

## Data Availability

The data presented in this study are available on request from the corresponding author.

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
