# Peer review of "In Search of the Perfect Triple BB Bond: Mechanical Tuning of the Host Molecular Trap for the Triple Bond B≡B Fragment"

_molecules, 2021, doi:10.3390/molecules26216428_

Round 1
Reviewer 1 Report
The manuscript by Zilberg and coworkers describes the computational study of B2 complexes with a series of bicyclic host compounds (III-VI). The authors demonstrate that a true BºB triple bond can be achieved through the elimination of π-back donation and the constrain of the linear NBBN fragment inside a rigid host. This is primarily a computational design study of appropriate host compounds. The proposed host structures are synthetically achievable. The authors have demonstrated that these bicyclic hosts can coordinate B2, stop the π-back donation and through mechanical strain, reinforce the triple bond.
The manuscript is well written and coherent. I believe the manuscript would be of interest to the broad readership of Molecules and I recommend publication once the following issues are addressed:
The introduction is appropriate and describes the state of art providing relevant references.
- The abstract is too long! Please shorten it.
- Line 35: “affording [2.2.1]bicyclic and a via..” remove the a before via
- Line 42: “that is a the” remove the a
- Line 43: “from the other side” should be replaced by “on the other side”
- Line 45: “the need for the keeping of linearity” replace by “the need to remain linear”
- Line 46: “Compounds with the B-B multiple bonds are still rare” remove “the” before B-B
- Line 61: “Later on the compound with a…” put a comma after later on
- Line 65: “CBBC fragment is a linear” remove a before linear
- Structures in scheme 2 have to be redrawn to be more clear.
- Line 148: where are the bond angles in table 2 to show the linearity of NBBN fragments.
- Line 163: “Host molecules III, IV and V have a different sizes” remove a before different sizes
- Line 173: “From other side, VII - hexa-oxo-173 icosane” should be replaced by “On the other side..”
- Line 188: “The addition of B2 to the host molecule III leads to the shortening of NN distance 188 from 6.048Å to 4.543Å due to.” This seems to be an incomplete sentence. Due to what?
- Line 194: “From other side…” should be replaced by “On the other side…”
- Line 201: “BB bond.16” reference 16 should be superscript.
- Line 202: “ 1720 to 1800 cm-1.24” reference 24 should be superscript.
- Line 211: “observed16 BB stretch” ref. 16 should be superscript.
- Lines 221-241 and 259-273: The font size has changed. Please make sure the font size remains the same throughout the manuscript.
- Line 275: “Producing of the perfect B≡B triple bond” should be replaced by “Productions of the perfect…”
- Line 276: “providing the preservation” should be replaced by “provides the preservation”
- Line 278: “Bicyclic host compounds IV-VII are satisfied to these requirements:” are satisfied should be changed to satisfy.
Author Response
We are grateful to the reviewer for his very helpful comments. We agree with him and accept the remarks.
Comments and Suggestions for Authors
The manuscript by Zilberg and coworkers describes the computational study of B2 complexes with a series of bicyclic host compounds (III-VI). The authors demonstrate that a true BºB triple bond can be achieved through the elimination of π-back donation and the constrain of the linear NBBN fragment inside a rigid host. This is primarily a computational design study of appropriate host compounds. The proposed host structures are synthetically achievable. The authors have demonstrated that these bicyclic hosts can coordinate B2, stop the π-back donation and through mechanical strain, reinforce the triple bond.
The manuscript is well written and coherent. I believe the manuscript would be of interest to the broad readership of Molecules and I recommend publication once the following issues are addressed:
The introduction is appropriate and describes the state of art providing relevant references.
- The abstract is too long! Please shorten it. Abstract corrected
- Line 35: “affording [2.2.1]bicyclic and a via..” remove the a before via corrected
- Line 42: “that is a the” remove the a corrected
- Line 43: “from the other side” should be replaced by “on the other side” corrected
- Line 45: “the need for the keeping of linearity” replace by “the need to remain linear” corrected
- Line 46: “Compounds with the B-B multiple bonds are still rare” remove “the” before B-B corrected
- Line 61: “Later on the compound with a…” put a comma after later on corrected
- Line 65: “CBBC fragment is a linear” remove a before linear corrected
- Structures in scheme 2 have to be redrawn to be more clear.
Scheme 2 redrawn. corrected
- Line 148: where are the bond angles in table 2 to show the linearity of NBBN fragments.
Correction: Polycyclic diboryne adducts III-B2÷VII-B2 have a linear NBBN fragment (calculations show very slight bending at the boron atoms NBB>178.5°) and a short BB bond.
- Line 163: “Host molecules III, IV and V have a different sizes” remove a before different sizes corrected
- Line 173: “From other side, VII - hexa-oxo-icosane” should be replaced by “On the other side..” corrected
- Line 188: “The addition of B2 to the host molecule III leads to the shortening of NN distance 188 from 6.048Å to 4.543Å due to.” This seems to be an incomplete sentence. Corrected: “The addition of B2 to the host molecule III leads to the shortening of NN distance 188 from 6.048Å to 4.543Å."
- Line 194: “From other side…” should be replaced by “On the other side…” corrected
- Line 201: “BB bond.16” reference 16 should be superscript. corrected
- Line 202: “ 1720 to 1800 cm-1.24” reference 24 should be superscript. corrected
- Line 211: “observed16 BB stretch” ref. 16 should be superscript. corrected
- Lines 221-241 and 259-273: The font size has changed. Please make sure the font size remains the same throughout the manuscript. corrected
- Line 275: “Producing of the perfect B≡B triple bond” should be replaced by “Productions of the perfect…” corrected
- Line 276: “providing the preservation” should be replaced by “provides the preservation” corrected
- Line 278: “Bicyclic host compounds IV-VII are satisfied to these requirements:” are satisfied should be changed to satisfy. corrected

Reviewer 2 Report
The Manuscript deals with the proposal of a models system to get a BB triple bond. Though the topic is interesting some concerns arise from the reading of the manuscript.
The manuscript was a little bit hard to read. Sometimes I got lost following the discussion. I’m not a native speaker, but a revision of English should be helpful for smoother reading.
The strategy proposed to get a triple bond is scientifically sound. Indeed, the A.A.s proposed the suppression of the p-backdonation and the design of a cage that allows retaining the linear NBBN geometry, to maintain (and to enhance) the s interaction along the NBBN axis to reach a triple bond interaction.
However, the A.A.s inferred the occurrence of the BB triple bond by focussing on the behaviour of structural parameters like BB and NN bond distances, and by comparison with experimental parameters of parents or similar systems. The conclusions are based on these grounds.
In my opinion, a methodological flaw in this manuscript relies upon the almost complete absence of a deep bond analysis. The only NBO analysis, for a system that could present unpaired electron is a quite rough approach, and the A.A.s themselves cite in the introduction:
“B2 fragment by two σ-donors L: should produce the structure with a triple bond L:→B≡B
←:L (I) between B-atoms. The trivial Lewis structure hides the fact that the ground state
of the parent B2 species is a triplet state 3Σg ̄ with pair of degenerate π-MO populated by
two electrons with parallel spin: (σ+) 2(σ-)2(πx)1( πy)1.5 The electronic configuration with
full π-MO: (σ+)2(σ-)0(πx)2( πy) 2 is a high lying excited singlet state 1Σg+.6,7 This singlet
state of B2 - 1Σg+ has a triple bond electronic configuration involving one σ-bond, two π-
bonds and two empty sp-hybrid orbitals at both B-atoms (see the MO interpretation in
Ref.8). “
Indeed, in ref.8 a more complete approach for the bond analysis was used, where: i) electrostatic, steric and orbital interactions were quantified as well as the s and p contributions, the main topic of this manuscript, to the orbital interactions. Nonetheless, a bond shortening could not depend only on a strengthening of the orbital interaction.
In my opinion, this is the main issue that needs to be addressed in this work, before publishing it on Molecules.
Further comments:
0. Keywords are missed
1. Introduction
i) the sentence:
“… .We propose to look for the answer to this problem through the elimination of the π back-donation effect that is the dominant factor of the weakening of π-components of the triple bond. ...”
must be moved at the end of the introduction.
ii) lines: 50 and 51. The references should be better formatted, i.e., the period at line 50 is written as superscript. The period is missed at line 51.
iii) A section with a description of the computational methods used in the manuscript is missed. Moreover, besides the computational protocols used, also how DEE (HM strain) and the DEE (dissociation) were calculated, must be reported here.
In addition, how the A.A.s estimated the Ionization Potential?
These pieces of information have been spread out all along the manuscript and should be gathered in ad hoc section.
2. Results and discussion
This section is still an overview of previous results. Perhaps a subsection with a more appropriate title should get clear the meaning of this paragraph.
However, is not clear to me where Fig 1 comes from? The models reported in Fig.1 were optimized by the A.A.s to show the distortion when p-backbonding lacks? This part seems to be a “frog jump” in the discussion.
3. Molecular design of compounds with the perfect triple BB bond
i) For a better comprehension of the hosts and the relative modified versions, IV and V should be oriented in the same way in Schemes 1 and 2.
ii) No pictures or data are reported concerning the NBO analysis for III-B2 up to VII-B2. The reader can’t get any idea about the NBO analysis, and the electronic structure of the systems. Which orbitals interact with which?
iii) lines 146, Supporting Information file is mentioned but I was not able to get it. This sentence should be removed if no Supplementary Information file is presented.
4. Strain energy of the host molecules
i) The sentence from lines 189 to 190 is incomplete.
ii) Table 2. DEE (HM strain) and the DEE (dissociation) should be given analogously. Why using 2 different units?
iii) reference at line 203 is not as superscript.
iv) lines 203,204: how the “Strengthening of the central BB fragments have been accompanied by a strong red-shift of the corresponding BB stretch mode up to2093 cm-1 for VII-B2. ...”?
should the strengthening of a bond and the ensuing shortening of the bond distance be usually followed by a blue shift?
Moreover, the Raman active frequency found by the authors is almost 300 cm-1 lower than the one for the N2 molecules (2300 cm-1). This is a huge difference that needs to be supported by further pieces of evidence to state the existence of a triple bond. Do the A.A.s have further explanations to support these conclusions?
Author Response
We are grateful to the reviewer for his comments and suggestions.
The main issue of our work is a design of diboryne derivatives without a π-back donation effect. The tuning of a host molecule to BB fragment and the calculation of structural and spectroscopic properties of the B2-host molecule complex were carried out for this purpose. We believe that the calculation and analysis of the experimentally observed characteristics are the focal point of the article. We agree with the reviewer that the bond analysis of cage effect could be a very useful approach. We decided to collect additional examples of complexes with a strong change of strain energy for the more complete bond order/charge distribution analysis. We are planning to include the computational data of the host-molecules I-VII and the corresponding diboryne adducts in our future studies.

Reviewer 3 Report
I have a few minor comments I would like the authors to address before the paper appears in print. The authors are right to notice the ‘mismatch’ of B2 and the amine ligands, that B2 is naturally 3Σ not 1Σ . And,oh yes. where does the singly bonded singlet,:B-B: energetically lie. This is not uncommon for multiple bonds, e.g. in HCCH we want 4∑CH not the ground state 2π for both fragments and in HCN, we want 4∑CH + 4S N. Indeed, are N2 and NO+ the only “perfect” triple bonds? There is also the “least motion” dimerization of CH2 to form C2H4, and of CF2 to form C2F4, in which CF2 is “naturally” used as its triplet excited state I think the authors should acknowledge these occurrence. Finally,on L.59 where the authors discuss OBBBBO and its ions they should distinguish between the experimental investigation of the monoanion, and the theoretical study of both the -1 and -2 species
Two other issues, the drawings for some of the authors’ species look crowded(cluttered) and occasionally incorrect, e.g. VI look slike there is a tetracoordinated ammonium nitrogen, and an dicoordinated.amide rather than two amines.
Author Response
We are grateful to the reviewer for his helpful comments. We agree with him and accept the remarks.
Comments and Suggestions for Authors
I have a few minor comments I would like the authors to address before the paper appears in print. The authors are right to notice the ‘mismatch’ of B2 and the amine ligands, that B2 is naturally 3Σ not 1Σ . And,oh yes. where does the singly bonded singlet,:B-B: energetically lie. This is not uncommon for multiple bonds, e.g. in HCCH we want 4∑CH not the ground state 2π for both fragments and in HCN, we want 4∑CH + 4S N. Indeed, are N2 and NO+ the only “perfect” triple bonds? There is also the “least motion” dimerization of CH2 to form C2H4, and of CF2 to form C2F4, in which CF2 is “naturally” used as its triplet excited state I think the authors should acknowledge these occurrence. The singly bonded singlet,:B-B: has a dominant configuration (2σg)2(2σu)2(3σg)2 and this is a high lying excited state according to MR-CISD data (Muller,TheorChemAcc 2001, 105, 227-243).
Finally,on L.59 where the authors discuss OBBBBO and its ions they should distinguish between the experimental investigation of the monoanion, and the theoretical study of both the -1 and -2 species corrected
Two other issues, the drawings for some of the authors’ species look crowded(cluttered) and occasionally incorrect, e.g. VI look like there is a tetracoordinated ammonium nitrogen, and an dicoordinated.amide rather than two amines. Scheme 2 redrawn. corrected

Reviewer 4 Report
My opinion is that this manuscript is interesting, it has been competently done and it is technically correct. Moreover, it is adequate for the readership of this journal. Therefore, I recommend publication after some minor revision as follows:
1) I think it should be indicated in the manuscript whether the calculated frequencies are harmonic or not, and whether a scaling factor was used.
2) there is no need to duplicate units in the header of the tables and in the first column.
3) for the convenience of the reader, the dissociation energy values in Table 2 should be given in kcal/mol.
4) I suggest to unify the notations for the functional: M06-2X or M062X.
5) ref 28. suggestion to cite gamess package as recommended by the authors with the package version listed: https://www.msg.chem.iastate.edu/gamess/citation.html.
5) my personal curiosity: I would like to know why the Authors chose cc-pvDZ bazis set instead of aug-cc-pVDZ for example or even pvTZ?
Author Response
We are grateful to the reviewer for his helpful comments. We agree with him and accept the remarks.
Comments and Suggestions for Authors
My opinion is that this manuscript is interesting, it has been competently done and it is technically correct. Moreover, it is adequate for the readership of this journal. Therefore, I recommend publication after some minor revision as follows:
- I think it should be indicated in the manuscript whether the calculated frequencies are harmonic or not, and whether a scaling factor was used.
Correction: p.5 "All structures were optimized, and harmonic vibrational frequencies were calculated."
2) there is no need to duplicate units in the header of the tables and in the first column. corrected
3) for the convenience of the reader, the dissociation energy values in Table 2 should be given in kcal/mol. corrected
4) I suggest to unify the notations for the functional: M06-2X or M062X. corrected
5) ref 28. suggestion to cite gamess package as recommended by the authors with the package version listed: https://www.msg.chem.iastate.edu/gamess/citation.html. corrected
5) my personal curiosity: I would like to know why the Authors chose cc-pvDZ bazis set instead of aug-cc-pVDZ for example or even pvTZ?
The calculations have been made for the different molecules including relatively big structures with adamantane groups. cc-pVDZ basis set was an optimal choice for the vibrational frequency calculations.

Round 2
Reviewer 2 Report
If the A.A.s can't perform the bond analysis in this manuscript, they must declare in the Conlcusions something like that: though all the outcomes in the manuscript support the occurrence of a triple a bond, the deepest bond analysis to quantify the sigma and pi contribution to the NBBN bonds is the subject of a future work.
Author Response
Thanks for your comments, I have revised and highlighted the rest of your comments both in the attachment and in the revised MS, please see the attachment.
